# First-principles quantum dynamical theory for the dissociative chemisorption of $H_2O$ on rigid Cu(111)

Zhaojun Zhang[1,*], Tianhui Liu[1,2,*], Bina Fu[1], Xueming Yang[1,2,3] & Dong H. Zhang[1,3]

Despite significant progress made in the past decades, it remains extremely challenging to investigate the dissociative chemisorption dynamics of molecular species on surfaces at a full-dimensional quantum mechanical level, in particular for polyatomic-surface reactions. Here we report, to the best of our knowledge, the first full-dimensional quantum dynamics study for the dissociative chemisorption of $H_2O$ on rigid Cu(111) with all the nine molecular degrees of freedom fully coupled, based on an accurate full-dimensional potential energy surface. The full-dimensional quantum mechanical reactivity provides the dynamics features with the highest accuracy, revealing that the excitations in vibrational modes of $H_2O$ are more efficacious than increasing the translational energy in promoting the reaction. The enhancement of the excitation in asymmetric stretch is the largest, but that of symmetric stretch becomes comparable at very low energies. The full-dimensional characterization also allows the investigation of the validity of previous reduced-dimensional and approximate dynamical models.

[1] State Key Laboratory of Molecular Reaction Dynamics and Center for Theoretical and Computational Chemistry, Dalian Institute of Chemical Physics, Chinese Academy of Sciences, Dalian 116023, China. [2] Department of Chemical Physics, University of Science and Technology of China, Hefei 230026, China. [3] Center for Advanced Chemical Physics and 2011 Frontier Center for Quantum Science and Technology, University of Science and Technology of China, Hefei 230026, China. * These authors contributed equally to this work. Correspondence and requests for materials should be addressed to B.F. (email: bina@dicp.ac.cn) or to D.H.Z. (email: zhangdh@dicp.ac.cn).

The molecular interactions and chemical reactions at metal surfaces are of great importance to many industrial applications. Dissociative chemisorption is one of the simplest surface chemical reactions, which undergoes the dissociation of a gas-phase molecule upon its collision with a surface, leading to chemisorbed products. It is a fundamental and often the rate-limiting step in many industrial heterogeneous catalytic processes, such as steam reforming, ammonia synthesis and the water–gas shift reaction[1]. Enormous progress has been obtained for the dissociative chemisorption dynamics in the past decades[2–18], due to its significance in surface chemistry and catalysis.

Thanks to the molecular beam and laser techniques, many pioneering experimental studies with quantum state resolution have been carried out, in which rich dynamics information, including mode specificity and bond selectivity in dissociative chemisorption processes was uncovered[3–7,12–16]. Owing to the involvement of a surface, the theoretical simulation for a gas-surface reaction is much more complicated than for a chemical reaction that occurs exclusively in the gas phase, even with small molecules. There has been intense effort to develop theory for reliable dynamics calculations of dissociative chemisorption of molecular species on surfaces. The purpose of this quest for quantitatively accurate theory is to provide a definitive understanding of the dynamics of dissociative chemisorption, which ultimately provides the basis for the control of chemical reactions at gas-surface interfaces and confident prediction of a model of heterogenous catalysis. It is gratifying that the system of diatomic molecules, dissociating on metal surfaces is sufficiently small to be amenable to full-dimensional (six-dimensional, 6D) quantum dynamics calculations on an accurate potential energy surface (PES)[8,19–32]. In particular for the dissociation of hydrogen molecule on metal surfaces, the state-of-the-art dynamics calculations, as well as the underlying chemical theory have been tested at a level close to that achieved for simple reactions in the gas phase[8,33].

The dissociative chemisorption of polyatomic molecules with multiple vibrational modes offers much richer complexity and mode specificity, thus much more attention has been attracted to them[34–36]. Despite significant progress made in the past two decades, it remains extremely challenging to investigate the dissociative chemisorption of polyatomic molecules on surfaces at a full-dimensional quantum mechanical level, due to the difficulties in constructing reliable high-dimensional PESs and developing practical quantum mechanical methodologies.

The classical approximation for nuclear motion has been an alternative to a full quantum characterization of dissociative chemisorption of polyatomic molecules. Although the *ab initio* molecular dynamics (AIMD) with energies computed on the fly, and quasiclassical trajectory calculations based on global PESs have shed valuable light on the dissociative chemisorption dynamics[37–41], both the two approaches neglect quantum effects and the former is also computationally expensive. It is worth noting that the effect of surface atomic motion can be taken into account with the *ab initio* molecular dynamics. An accurate full quantum characterization of dissociative chemisorption of polyatomic molecules is highly desirable in view of potentially important quantum effects, such as tunnelling, zero-point energy (ZPE) and resonances.

The dissociative chemisorption of water on transition-metal surfaces is of fundamental importance in understanding many industrial heterogeneous catalytic processes, such as steam reforming, and represents the rate-limiting step in low-temperature water–gas shift reaction on copper catalysts[42,43]. A total of nine degrees of freedom should be considered on a rigid surface, rendering it difficult to develop an accurate, global PES

and formidable to carry out a fully coupled nine-dimensional (9D) quantum dynamics calculations. Previous studies of water on surfaces resorted to approximate or reduced-dimensional quantum models.

An approximate approach relying on a reaction path Hamiltonian[44], markedly simplifies the polyatomic problem. Farjamnia and Jackson[45] employed the reaction path Hamiltonian approach to investigate the dissociative chemisorption of water on Ni(111), which treats all the internal motions of water using a vibrationally adiabatic basis set, and produces the minimum-barrier-site reaction probability from wave packet propagations. Sudden models were used to average over the impact sites and lattice motion[46].

Development of the reduced-dimensional quantum approaches is prevalent in understanding the dissociation of water on metal surfaces, starting from the simplest pseudo-diatomic molecule model by Tiwari and co-workers[47]. Guo and co-workers performed 6D quantum dynamics calculations to study the mode specificity of $H_2O$ and bond selectivity of HOD on a rigid flat Cu(111) surface[48–51], employing their 6D PESs developed by permutationally invariant polynomials (PIPs)[52,53]. The 6D model neglects the effects of impact sites and surface corrugation. The obtained strong mode specificity can be rationalized by recently proposed sudden vector projection model[54,55].

The first quantum-state-resolved molecular beam experiment on the dissociative chemisorption of $D_2O$ on Ni(111) demonstrated a large enhancement in reactivity upon excitation of the asymmetric stretch of $D_2O$ (ref. 16). The observed mode specificity was semiquantitatively understood by Guo and Beck and co-workers[16] using the 6D quantum dynamical approach, where the effects of impact sites and lattice motion were approximated using the sudden models proposed by Jackson and co-workers[39,46,56–58]. Recently, Jiang and Guo[51] investigated 6D site-specific dissociation probabilities of this system, but based on a new 9D PES developed by PIP-neural network (PIP-NN) method.

Very recently, we reported the first seven-dimensional (7D) quantum dynamics study for the dissociative chemisorption of $H_2O$ on Cu(111), based on an accurate 9D PES developed by NN approach[18,59]. The dissociation probabilities exhibit strong azimuthal angle dependence, indicating the 6D quantum model neglecting the azimuthal angle can introduce substantial errors.

The new 7D quantum dynamics study for the title gas-surface reaction has provided new insight into the dissociative chemisorption dynamics[18,59]. Nevertheless, it is still a provisional model with reduced-dimensional approximation, in which the surface lateral coordinates are fixed in a specific site of the impact. A quantitatively accurate description of the title gas-surface reaction can only be verified by comparison with a fully coupled 9D quantum mechanical approach based on an accurate global PES.

In this work, we report the full-dimensional quantum dynamics study of the dissociative chemisorption of $H_2O$ on a rigid Cu(111) surface, allowing all the nine molecular degrees of freedom to be fully coupled. The challenging full-dimensional quantum mechanical calculations not only offer the dynamics features with the highest accuracy, but also allow a conclusive examination of previous dynamical approximations.

## Results

**9D dissociation probabilities.** The full-dimensional (9D) dissociation probability for $H_2O$ initially in the ground vibrational state, together with the 7D results on the fixed top, bridge, fcc, hcp and transition-state (TS) sites are illustrated in Fig. 1 in the kinetic energy region of (0.9, 1.8) eV. Due to the non-reactive of

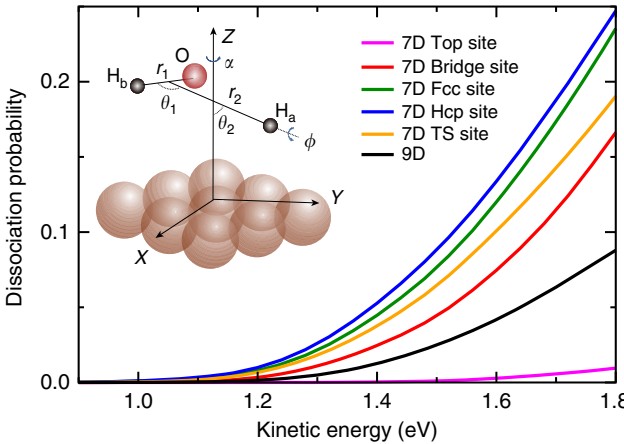

**Figure 1 | Full-dimensional (9D) and seven-dimensional (7D) site-specific results.** The 9D dissociation probability and 7D site-specific (top, bridge, fcc, hcp and TS sites) probabilities with $H_2O$ initially in the ground rovibrational state (000). The 9D molecular coordinates of the $H_2O/Cu(111)$ system are shown in the inset.

one OH bond in current theory, the computed dissociation probabilities were multiplied by a factor of 2 that correspond to the results for two reactive bonds. On the whole, all the dissociation probabilities are smooth functions of the kinetic energy, and increase monotonically with increasing kinetic energy. As has been discussed before[29,30,41,51,59], the dissociation probability for $H_2O$ fixed at the top site is extremely small, due to the topography of the PES and dynamical process, but not the static barrier height. Significant differences between the 9D dissociation probability and the results calculated by reduced-dimensional (7D) fixed-site approach are seen. The 9D probability is substantially smaller or larger than the 7D fixed-site probabilities in the entire energy region. At the kinetic energy of 1.8 eV, the 9D probability is larger than the probability on the fixed top site by a factor of over 8, but smaller than the rest of four fixed-site results by a factor of roughly 1–2. In addition, these factors turn to much larger with the decreasing kinetic energy. Obviously, the 7D quantum model neglecting the surface lateral coordinates cannot produce quantitatively accurate dissociation probabilities and the full-dimensional calculations are remarkably important to characterize the dynamics.

**Vibrational excitations.** As the water reactant has three vibrational modes, including the symmetric stretching ($\nu_1$), bending ($\nu_2$) and asymmetric stretching ($\nu_3$) modes, respectively, various vibrational excitations of $H_2O(\nu_1\nu_2\nu_3)$ should have different influences on its interaction with metal surfaces. We are now capable of gaining insight into the mode specificity of the title gas-surface reaction at a full-dimensional quantum mechanical level, which is significant in the reactivity control and an in-depth understanding of the dynamics.

Figure 2 shows the 9D dissociation probabilities for $H_2O$ initially in six lowest non-rotating vibrational states in both linear and log scales. The results are compared as a function of total energy, which is measured with respect to the asymptotic energy of ground rovibrational state of $H_2O(000)$. As seen in Fig. 2a, the dissociation probabilities of five vibrationally excited states are all larger than that of the ground state in the entire energy region, implying that excitations in vibrational modes of $H_2O$ enhance the reactivity for the dissociative chemisorption of $H_2O$ on Cu(111). The reactivity enhancement by exciting the stretching modes is consistent with the Polanyi's rules for a late-barrier reaction, and the enhancement due to the bending mode is

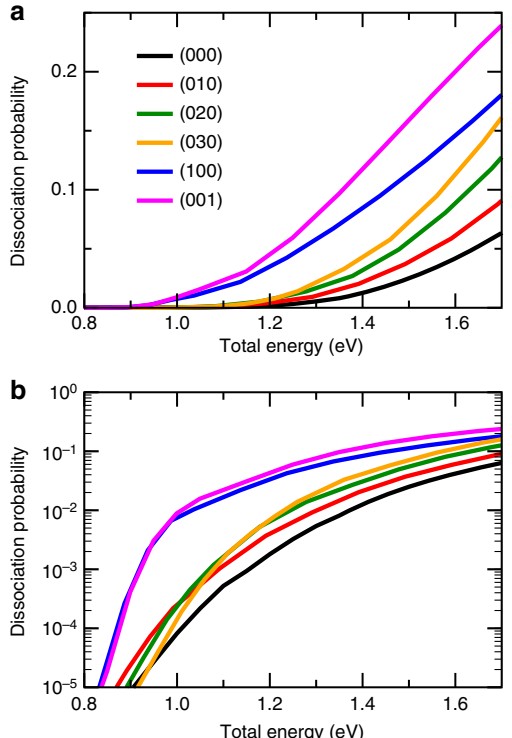

**Figure 2 | The reactivity enhancement by vibrational excitations of reactant.** (**a**) The full-dimensional dissociation probabilities with $H_2O$ in six non-rotating vibrational states as a function of the total energy. (**b**) Same as **a**, except in a logarithmic scale.

evidenced by the fact that the H–O–H angle of the saddle point (119.21°) is larger than that in isolated $H_2O$ (104.41°). The three bending overtones (010), (020) and (030) enhance the dissociation probability, but with smaller degrees of enhancement as compared with the stretching modes, although the excitation energy of the (030) state is even larger than that of the asymmetric stretch (001). Vibrational excitations in the stretching modes are more efficient than excitations in the bending modes, due to the stronger couplings of $H_2O$ stretch with the reaction coordinate. It is interesting to find that the enhancement of excitation in the asymmetric stretching mode is larger than that of the symmetric stretching mode (100), consistent with the performance in 7D[59], but not in the 6D calculations[50], where the larger enhancement of symmetric stretching mode is observed.

From the corresponding dissociation probabilities shown in a logarithmic scale in Fig. 2b, we can obviously see that in the very low-energy regions below the classical barrier (1.08 eV) there exist considerable dissociation probabilities for the ground vibrational state of $H_2O$. The ZPEs of reactant asymptote and TS are 0.57 and 0.43 eV, respectively, so that the ZPE-corrected barrier height is 0.94 eV. Thus, the reactivity below the classical barrier (down to roughly 0.9 eV) is due to both the ZPE and tunnelling effects. In addition, the overall reactivity enhancement from vibrational excitations increases as the total energy decreases. The enhancement factors of the reactivity relative to the ground state at the total energy of 1.1 eV are 2.98, 5.52, 4.47, 34.26 and 53.99 for the (010), (020), (030), (100) and (001) states, respectively, which are much larger than those of 1.42, 2.02, 2.57, 3.04 and 3.81 at the total energy of 1.7 eV, respectively. The largest enhancement of excitation in the asymmetric stretching mode is presumably due to the high excitation energy of this mode and its strong couplings with the reaction coordinate. We also note that the enhancement of excitation in the symmetric stretching mode

**Table 1 | Vibrational efficacies for five non-rotating excited vibrational states in dissociative chemisorption of $H_2O$ on Cu(111) calculated by full-dimensional quantum dynamical approach.**

| Vibrational state | Vibrational energy (eV) | Vibrational efficacy, $\eta$ | | | | |
|---|---|---|---|---|---|---|
| | | $P_0 = 10^{-5}$ | $P_0 = 10^{-4}$ | $P_0 = 10^{-3}$ | $P_0 = 10^{-2}$ | $P_0 = 0.05$ |
| (010) | 0.192 | 1.18 (1.05)* | 1.29 | 1.34 | 1.39 | 1.43 |
| (020) | 0.378 | 1.04 (0.884)* | 1.12 | 1.22 | 1.35 | 1.42 |
| (100) | 0.437 | 1.17 (0.999)* | 1.32 | 1.54 | 1.79 | 1.84 |
| (001) | 0.449 | 1.16 (0.932)* | 1.30 | 1.52 | 1.81 | 1.93 |
| (030) | 0.560 | 0.98 | 1.04 | 1.13 | 1.26 | 1.37 |

*The numbers in parentheses are the 6D results for the vibrational efficacy in ref. 50.

become comparable to the asymmetric stretching mode at very low energies ($<1.0\,eV$).

The relative enhancement is often quantified in terms of the vibrational efficacy, which is described as $\eta = [E_i(0, P_0) - E_i(v, P_0)]/(E_v - E_0)$, where $E_i(0, P_0)$ and $E_i(v, P_0)$ are the kinetic energy for the vibrational ground (0) and excited (v) states to give the same dissociation probability $P_0$, while $E_0$ and $E_v$ are the corresponding vibrational energies[45,50]. The computed results are shown in Table 1. The values of $\eta$ are always larger than unity, indicating excitations in all the vibrational modes are more efficacious than the same amount of translational energy in promoting the reaction. The vibrational efficacy increases with the increase of dissociation probabilities and collision energies. Besides, we see much larger values of vibrational efficacy, that is, much stronger mode specificity in the present 9D work than those obtained by the 6D quantum model[50].

**Validity of site-averaging approximations.** A full-dimensional quantum dynamical characterization of a polyatomic molecule-surface reaction is ideal, which is often many theoreticians strive for, due to the fact that it can provide more reliable results and a quantitatively accurate understanding of the reaction dynamics. Nevertheless, such calculations for polyatomic dissociations, such as an explicit treatment of the 15 degrees of freedom for rigid $CH_4$, are still formidable. Employing the reduced-dimensional dynamical model, as well as the impact site approximation to accurately approximate the full-dimensional quantum mechanical reactivity should be a reliable and promising approach. The current full-dimensional quantum dynamics study for $H_2O$/Cu(111) gives a good opportunity to validate two existing implementations of site-averaging approximations for polyatomic molecule-surface reactions.

The site-sudden approximation is proposed by Jackson and Nave[46] and also extensively used[16,39,56,60], which characterizes the dynamics with the assumption that dissociation probabilities at different impact sites have the same energy dependence as that at one single site, for example, TS site, but varying with the static barrier height. With that approximation, the potentials at various impact sites are approximated by a harmonic expansion around one specific site.

Recent dynamics calculations demonstrated that the reactivity mainly depends on the topography of the PES rather than the static barrier height[29,30,32,41,51], indicating the site-averaging approximation with harmonic potential (SAHP) employed in polyatomic molecule-surface reactions might not be quite reliable[16,39,46,56,60]. We recently proposed and validated a new site-averaging approximation with exact potential (SAEP) in diatomic molecule-surface reactions[29–31], where the full-dimensional reactivity was well reproduced by averaging reduced-dimensional fixed-site results over multiple impact sites with appropriate relative weights. As in the following, both the two implementations of site-averaging approximations are first

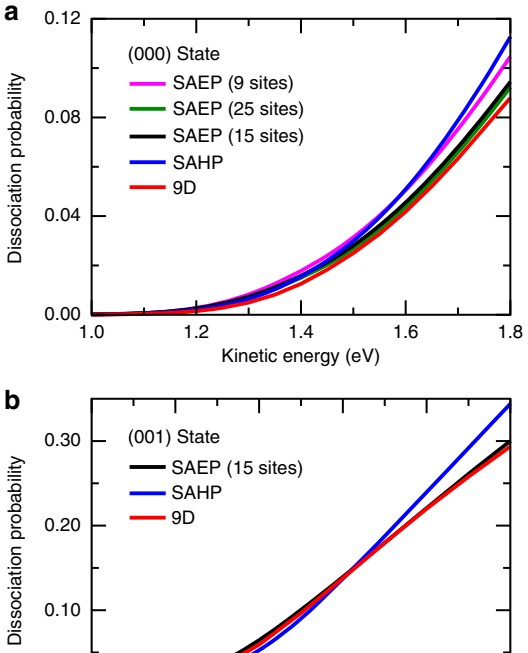

**Figure 3 | 9D and site-averaging 7D results. (a)** Comparisons of dissociation probabilities obtained by averaging the 7D results over 9, 25 and 15 impact sites with appropriate relative weights (SAEP), the probability obtained by averaging over the results for many $(X,Y)$ sites using SAHP, and the full-dimensional dissociation probability, for $H_2O$ initially in the (000) state. **(b)** The dissociation probabilities obtained by the SAEP and SAHP approximations, together with the full-dimensional dissociation probability, for $H_2O$ initially in the (001) state.

rigorously examined in the dissociative chemisorption of water on metal surfaces.

The dissociation probabilities from the two implementations of site-averaging approximations (SAEP and SAHP) are compared with the 9D dissociation probability in Fig. 3. First, the SAEP results by averaging 7D dissociation probabilities over 9 sites and 25 sites in the $C_{3v}$ unit cell of Cu(111), with appropriate relative weights were calculated for $H_2O$ initially in the ground vibrational state (000; see Supplementary Fig. 1 and Supplementary Discussion for the schematic of the distribution of 9 sites and 25 sites, and the detailed site-averaging approach). As shown in Fig. 3a, the overall behaviour of the nine site-averaging dissociation probability resembles that of the 9D dissociation probability, although the magnitude of the former is larger than the latter particularly at high kinetic energies. The agreement

between the 25 site-averaging probability using the SAEP and 9D probability is much better and the differences between them are negligible, indicating the validity of SAEP is confirmed again in the title polyatomic molecule-surface reaction.

As the static barrier heights and 7D dissociation probabilities of the hcp and fcc sites (Fig. 1) are quite similar, we assume the

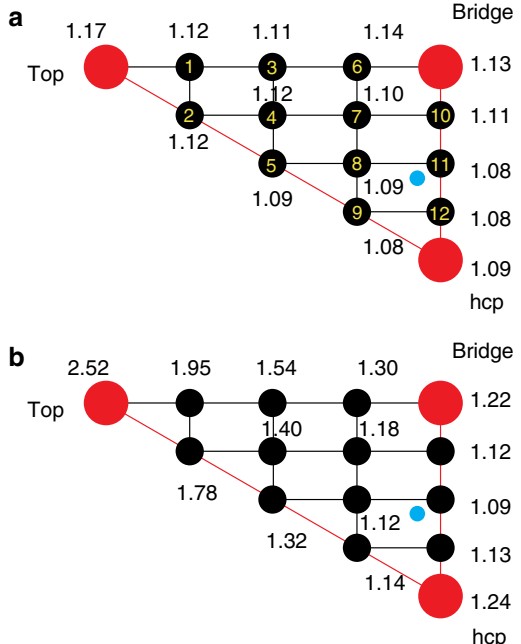

**Figure 4 | Static barrier heights.** (**a**) Static barrier heights (in eV) obtained on the exact fitted PES for 15 impact sites in the irreducible triangle of Cu(111) within the approximate $C_{6v}$ symmetry. The site numbers are also labelled and the TS site is indicated by a blue circle. (**b**) Same as **a**, except by the HP approximation.

$C_{6v}$ symmetry of the Cu(111) surface, where the fcc and hcp sites are indistinguishable on the PES. As a result, the 25 sites included in the averaging can be reduced to 15 sites further (one-half of the triangle), as shown in Fig. 4. The 15 site-averaging probability agree quite well with the 25 site-averaging probability, as well as the 9D result, indicating the full-dimensional probability of (000) state can be accurately reproduced by averaging 7D dissociation probabilities over 15 sites.

Following Jackson and Nave[46], as detailed in the Supplementary Discussion, we obtained the dissociation probability averaging over all important impact sites with SAHP. We can see in Fig. 3a that the dissociation probability with SAHP is close to the 15/25 site-averaging probability with SAEP and 9D probability in the low-energy region, but larger than them at kinetic energies >1.5 eV.

To further validate the two implementations of site-averaging approximations, we also compare the SAEP, SAHP and 9D probability curves for the vibrational excitation in asymmetric stretching mode of $H_2O$ (001) in Fig. 3b. The agreement between the 15 site-averaging probability with SAEP and 9D probability is excellent and impressive, which further confirms the validity of our SAEP. However, the probability with SAHP is slightly smaller at low kinetic energies, but much larger at kinetic energies >1.0 eV than the 9D result for the (001) state. Those discrepancies imply the SAHP might introduce some errors in describing the full-dimensional quantum mechanical reactivity.

To give a more explicit comparison between the two implementations of site-averaging approximations, the exact static barrier heights on the PES and those obtained using the HP approximation at 15 impact sites mapped in the irreducible triangle within the approximate $C_{6v}$ symmetry are illustrated in Fig. 4a,b, respectively. Obviously, the site-specific barrier height obtained by HP is always larger than the exact barrier height on the PES. More significant differences are seen when the distance between a specific site $(X,Y)$ and the TS site increases. For example, the barrier height with HP at the hcp site (1.24 eV) is 0.15 eV larger than the exact barrier height (1.09 eV) on the PES,

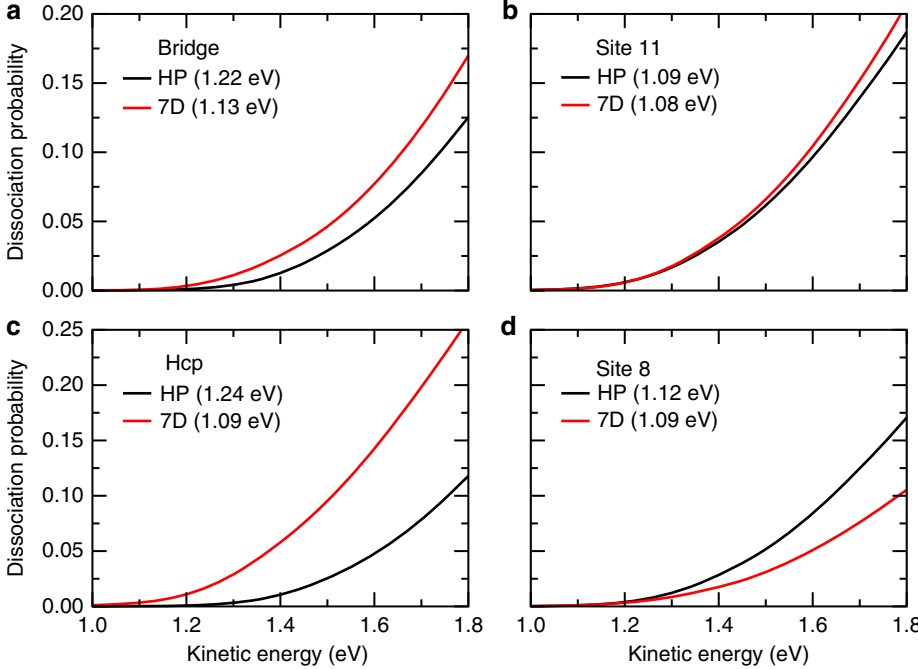

**Figure 5 | Site-specific dissociation probabilities.** (**a–d**) The site-specific dissociation probabilities obtained by the HP approximation and those calculated by 7D quantum dynamics calculations at the fixed bridge site, site 11, hcp site and site 8, respectively, with $H_2O$ initially in the ground rovibrational state. The barrier heights obtained from the HP approximation and the exact PES are also indicated in the parentheses.

while at the top site the former (2.52 eV) is 1.35 eV larger than the latter (1.17 eV). These results indicate that the HP expansion cannot be validated for the barrier corrugation. The potential expanded in a harmonic approximation around a fixed site can be too high, which is expected in general and was discussed by Jackson et al.[39]

Furthermore, the site-specific dissociation probabilities obtained by HP and by the 7D quantum dynamics calculations for the (000) state of $H_2O$ at 15 impact sites discussed above are calculated and compared. The results at four impact sites, that is, the bridge site, site 11, hcp site and site 8, as shown in Fig. 4a are illustrated in Fig. 5. Those results at rest of 10 sites are presented in Supplementary Figs 2–4, except for the top site, where the probability obtained by HP in the energy region considered are completely zero, due to the very high barrier height (2.52 eV) from HP. Overall, significant differences are observed between the two batches of results in the entire energy region, indicating the HP approximation can give rise to substantial errors in attaining the 7D site-specific dissociation probabilities. The farther this specific site is away from the TS site, the more remarkable differences are seen between the two batches of results. Although we did not see such significant differences in Fig. 3 after averaging many site-specific dissociation probabilities, it can be fortunate because the positive and negative errors induced by the HP approximation balanced out in the averaging.

## Discussion

We have carried out the full-dimensional quantum dynamics study for the dissociative chemisorption of $H_2O$ on a rigid Cu(111) surface, which rigorously treats and fully couples all the nine molecular degrees of freedom, to achieve a quantitatively accurate understanding of this centrally important gas-surface reaction. An accurate, full-dimensional PES constructed recently by the NN fitting to density functional theory (DFT) energy points was employed. The full-dimensional model allows the surface corrugation, as well as the translational movement of the $H_2O$ moiety. Large differences between the full-dimensional (9D) and site-specific 7D reactivity are observed, indicating the significance of the challenging full-dimensional quantum mechanical calculations. The excitations in five lowest vibrational modes greatly promotes the reaction, which are more efficient than increasing the translational energy. The enhancement of the excitation in the asymmetric stretching mode (001) is largest, to which that of the symmetric stretch (100) become comparable at very low energies, whereas that of the bending mode (010) is smallest, presumably due to the strong couplings of $H_2O$ stretch with the reaction coordinate. The bending excited state (030) has the highest vibrational energy, but gives rise to the smallest vibrational efficacy. The full-dimensional quantum dynamics study also allows the investigation of the validity of two implementations of site-averaging approximations. It is found that the full-dimensional reactivity can be accurately reproduced by averaging site-specific 7D results from quantum mechanical calculations over 15 impact sites, validating our SAEP in this polyatomic molecule-surface reaction. In contrast, the SAHP can introduce substantial errors in obtaining the site-specific dissociation probabilities. The resulting reactivity from SAHP may not accurately approximate the full-dimensional reactivity. The validity of SAEP observed in diatomic molecule-surface reactions and the $H_2O$/Cu(111) reaction should generally hold in many gas-surface reactions. It probably provides a good opportunity to investigate the dissociative chemisorption dynamics of polyatomic molecules on metal surfaces at a full-dimensional quantum mechanical level by site-averaging the reduced-dimensional results over multiple impact sites.

The surface lattice motion should also be considered, so that a direct comparison with the experimental data can be achieved.

## Methods

**9D PES.** The 9D PES was constructed by using the NN fitting to a total of 81102 DFT energy points[18]. This fit results in an overall very small root mean square error of only 9.0 meV, but is significantly smaller (6.0 meV) for energy points < 2.0 eV relative to the $H_2O + Cu(111)$ asymptote, representing the unprecedented fitting accuracy for PESs of polyatomic-surface reactions. Readers can refer to Supplementary Methods for the details of 9D PES.

**9D time-dependent wave packet approach.** A total of nine degrees of freedom should be considered for the fully coupled quantum dynamics calculations of the dissociative chemisorption of $H_2O$ on a corrugated, rigid metal surface. The 9D Hamiltonian and time-dependent wave function for the title reaction is expressed in terms of molecule coordinates ($x, y, Z, r_1, r_2, \theta_1, \theta_2, \phi, \alpha$ ; shown in Supplementary Fig. 5). Details of quantum dynamical methodology are given in the Supplementary Methods section and the convergence of quantum results with respect to numerical parameters is shown in Supplementary Figs 6 and 7.

**Data availability.** The data that support the findings of this study are available from the corresponding author upon request.

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

## Acknowledgements

This work was supported by the National Natural Science Foundation of China (grant nos 91421315, 21433009, 21590804 and 21303197), the Ministry of Science and Technology of China (2013CB834601), the Youth Innovation Promotion Association (2015143) and the Chinese Academy of Sciences.

## Author contributions

D.H.Z., X.Y. and B.F. conceived and supervised the research; Z.Z. and T.L. performed the research; Z.Z., T.L., B.F. and D.H.Z. analysed the data; and B.F. and D.H.Z. wrote the manuscript.

## Additional information

**Competing financial interests:** The authors declare no competing financial interests.

