## [Peer Review File · Nature Communications]

Reviewers' comments:

Reviewer #2 (Remarks to the Author):

This is a superb article on the quantum dynamics of dissociative chemisorption of a polyatomic molecule on a rigid surface. All nine molecular degrees of freedom are included in the model, which offers a global and accurate picture of the dissociation dynamics. The nine-dimensional potential energy surface was constructed in a previous study by the same group by fitting many DFT points. The nine-dimensional quantum dynamical results allowed (a) a quantitative characterization of a complex chemical reaction dynamics at the gas-surface interface, and (b) definitive testing of reduced dimensional models. It represents an important milestone in our understanding of surface reaction dynamics.

The results confirmed the conclusion of previous reduced dimensional studies of the same process, that all vibrational modes of H₂O enhance reactivity more than the translational excitation. It also revealed errors introduced by reduced dimensional models.

The comparison with reduced dimensional models suggests that the site-average model is generally valid, which is good news that the dynamics can be reasonably described with lower dimensional models.

Overall, the manuscript was written clearly and the results are important and impactful. I strongly recommend its publication after the authors address the following minor issues:

1. It is important that the title and main text give the reader a clear idea that the surface lattice motion is not included in the model. This is an important approximation that needs to be stated clearly. I recommend the addition of the word "rigid" before Cu(111) in the title and throughout the manuscript. In addition, the authors should state in the paper clearly that the neglect of the surface atomic motion is the main reason that a direct comparison with the experimental data is not yet possible.
2. The word "full dimension" can be misleading, because no surface atom is included in the model. I would suggest the use of 9D, rather than full dimension.
3. There have been several important reviews on the theoretical studies of dissociative chemisorption of polyatomic molecules (J. Phys. Chem. A 2014, 118, 9615-9631. and Chem. Soc. Rev. 2016, in press (DOI: 10.1039/c5cs00360a)). These reviews should be cited in the introduction.

Reviewer #3 (Remarks to the Author):

This paper reports what is, to my knowledge, the first quantum dynamical study of the dissociative chemisorption of a triatomic molecule on a metal surface in which all molecular degrees of freedom are treated essentially without approximation (i.e., full-dimensional with respect to the number of degrees of freedom). This is a formidable achievement, as nine degrees of freedom have to be described with quantum dynamics, and well deserving of publication in a journal like Nature Communications. The results presented can serve as a benchmark for calculations testing the quality of more approximate treatments.

Overall, the level of the writing is quite high. For instance, the paper comes with a sound introduction, which describes the state-of-the-art of the field up to the present paper quite well. Nevertheless, the paper can still be improved on a number of important points. I suggest the authors to make the following changes, after the implementation of which the paper can be published. I have listed quite a few changes, but this merely reflects my enthusiasm for the paper.

Major points:

1. Title: I suggest to change the title to: "First principles quantum dynamical theory ...". The reasoning underlying this: The dynamics is first principles, but DFT at the GGA level is at best considered "non-empirical", because the selection of any GGA functional implies choices, even if the functional is based on non-empirical constraints.
2. In line 6 of page 3, in the statement concerning AIMD "... is also computationally too expensive." the word "too" should be removed, its use is too normative. "Too expensive" compared to what? Is not nine-dimensional quantum dynamics even more computationally expensive? I also suggest that the authors mention that an advantage of AIMD would be that the effect of the motion of the surface atoms (and therefore of surface temperature) can be taken into account with no extra cost.
3. Concerning the last sentence of page 5: The comparison in Figure 1 only makes it clear that motion along the surface needs to be treated. It does not yet make it clear that it has to be done in a fully coupled way, an approximate treatment (site-sudden) could well do the job. This distinction needs to be made clear.
4. Top page 6, concerning vibrational enhancements: Polanyi's rules only say something about how stretch modes might enhance the reactivity, but the 2 authors mention these rules also in the context of vibrational enhancement of reaction by exciting the bending modes. Authors please make this distinction, possibly referring to the more general model of Hua Guo.
5. Regarding the second paragraph on page 6: it would be good to include a Table stating not only the minimum (classical) barrier height in the PES, but also the zero-point energy corrected barrier height in the paper. This is crucial information. For instance, as the authors should also discuss in the paper, reaction below the classical barrier does not need to be indicative of tunneling effects, it could also be due to zero-point energy effects (if the zpe at the barrier is lower than the zpe of the reactants).
6. I suggest to extend Table 1 with one column, showing 6D results for the vibrational efficacy for one value of the reaction probability. This seems important in the light of the first paragraph on page 7, it would be better not to have to look up reference 46 to believe the statement made about the comparison between the 6D and 9D results.
7. I suggest that the authors change their terminology regarding the "two site approximations" on page 7, now called "site-averaging" and "site-sudden". Both approximations are actually of the "site-sudden", or "site-averaging" type, they have just been implemented differently. Specifically, in one case the exact fitted potential is used, in the other case an approximation to it, using a harmonic expansion around one specific site. The terminology should be changed to reflect this, for instance "site-averaged exact potential" and "site averaged harmonic potential" with acronyms SAEP and SAHP. Also, I suggest to drop any mention of "two site approximations", it might be better to say "two implementations of site-averaging approximations", or something along these lines.
8. Page 8, regarding the difference between "site-sudden" and "site-averaged": far away from the site around which the potential is expanded in a harmonic approximation, the

expanded potential becomes too high. This was also invoked to explain AIMD and site-sudden results for methane + Ni(111), see Jackson et al., JCP141, 054102, 2014. The authors might discuss whether this problem is too to be expected in general if an harmonic expansion around a fixed site is used.

9. In papers like these, the reaction probability is computed by calculating the probability that one of the OH bonds breaks. The correct reaction probability is only obtained if the computed "reactive flux" is multiplied by a factor two. If this was indeed done, this should be explicitly indicated in both the paper and the SI. If this was not yet done, all plots should be changed with reaction 3 probabilities multiplied by a factor two, and again this should be indicated in the paper and the SI.

10. In the SI, I do not see the estimated convergence of the 9D QD calculations. Can the authors add results of convergence tests? This should be less difficult than it might seem. Convergence w.r.t. rotational basis set can be tested in calculations using a fixed site approximation. Convergence w.r.t. numbers of points in X and Y could be tested in a calculation where approximations are made to one or two of the rotations and/or the bond not allowed to be broken. The number of points in X and Y (7 x 7) would seem to be rather low in the 9D calculations, of the same order as in the "site-averaged" calculations (8 x 8 with 25 points). The question could then arise whether the 9D calculation represents a true benchmark.

Minor points:

a) Page 2: "...attracted to them."

b) In line 9 of page 4 I suggest to talk of the work of "Guo and Beck and coworkers", i.e. mention both senior scientists on the paper.

c) On page 4 I suggest to say in line 3 from the bottom: "The quantitatively accurate description of the title gas-surface reaction can only be verified by comparison with a fully coupled 9D ... ". In other words, approximative treatments might exhibit quantitative accuracy, but to be sure the benchmark full-dimensional (w.r.t. the molecular degrees of freedom) calculation needs to be done first.

d) First two lines page 5: "... 9 degrees of freedom to be fully coupled."

e) Reference 21 is wrong. It should be "Dai, JQ" and not "Dai, J". Authors please check all citations on the Web of Science, especially the ones with Chinese authors in them.

f) Regarding reference 22: the better paper to cite is Kroes, Baerends, Mowrey, Phys.Rev.Lett.78, 3583, 1997. This paper is the first 6D quantum dynamics paper on fully activated dissociation of H₂, and has both an earlier "received" and "accepted" date than Ref.21.

g) A recommendation for future work is to use PBE instead of PW91. PBE was designed to supersede PW91, see the PRL 1996 of Perdew, Burke, and Enzerhof.

h) The SI should state a measure of the accuracy with which the PES was fitted (for instance, a RMSD for data not in the training set).

i) In the SI on page 3, line 5 should mention that the expansion is also in Fourier basis functions of X and Y. 4

j) Page 5 SI: line 5 paragraph 3: should read "top, hcp, and fcc sites". I.e., replace "hollow" by "hcp". Same elsewhere.

k) Bottom page 6 SI: the readability of the SI can be tremendously improved by making reference to Fig.4a of the main paper for the numbering of the sites. Authors: many readers will read the paper in the order Intro, SI, rest of paper.

l) The caption of Fig.S3 should likewise refer to Fig.4a of the main paper.

This concludes my suggestions.

We thank the referees for their careful reading and useful comments. The detailed revisions to the manuscript are presented as follows. All the referees' comments are reproduced before the responses (indicated by blue fonts).

For Referee #2:

1. It is important that the title and main text give the reader a clear idea that the surface lattice motion is not included in the model. This is an important approximation that needs to be stated clearly. I recommend the addition of the word "rigid" before Cu(111) in the title and throughout the manuscript. In addition, the authors should state in the paper clearly that the neglect of the surface atomic motion is the main reason that a direct comparison with the experimental data is not yet possible.

We've added "rigid" in the title and throughout the main text when necessary. We've also added the following sentence at the end of the main text: "The surface lattice motion should also be considered so that a direct comparison with the experimental data can be achieved."

2. The word "full dimension" can be misleading, because no surface atom is included in the model. I would suggest the use of 9D, rather than full dimension.

We've indicated in the abstract and main text that "a total of all 9 molecular degrees of freedom fully coupled" and also "a rigid Cu(111) surface", so that it is clear to readers that the full-dimension corresponds to the molecular degrees of freedom.

3. There have been several important reviews on the theoretical studies of dissociative chemisorption of polyatomic molecules (J. Phys. Chem. A 2014, 118, 9615-9631).

and Chem. Soc. Rev. 2016, in press (DOI: 10.1039/c5cs00360a). These reviews should be cited in the introduction.

The two reviews have been cited in the Introduction.

For Referee #3:

Major Points:

1. Title: I suggest to change the title to: "First principles quantum dynamical theory ..." . The reasoning underlying this: The dynamics is first principles but DFT at the GGA level is at best considered "non-empirical", because the selection of any GGA functional implies choices, even if the functional is based on non-empirical constraints.

We've changed the title to "First-principles quantum dynamical theory for the dissociative chemisorption of H₂O on rigid Cu(111)".

2. In line 6 of page 3, in the statement concerning AIMD "... is also computationally too expensive." the word "too" should be removed, its use is too normative. "Too expensive" compared to what? Is not nine-dimensional quantum dynamics even more computationally expensive? I also suggest that the authors mention that an advantage of AIMD would be that the effect of the motion of the surface atoms (and therefore of surface temperature) can be taken into account with no extra cost.

In line 6 of page 3, we've removed the word "too" before "expensive". In addition, we've added one sentence "It is worth noting that the effect of surface atomic motion can be taken into account with the AIMD." in the same paragraph.

3. Concerning the last sentence of page 5: The comparison in Figure 1 only makes it clear that motion along the surface needs to be treated. It does not yet make it

clear that it has to be done in a fully coupled way, an approximate treatment (site-sudden) could well do the job. This distinction needs to be made clear.

We can see in Fig. 3 that the differences between the 9D and site-sudden results are substantial, in particular in the high kinetic energy region. As a result, the fully coupled 9D treatment is essential.

4. Top page 6, concerning vibrational enhancements: Polanyi's rules only say something about how stretch modes might enhance the reactivity, but the authors mention these rules also in the context of vibrational enhancement of reaction by exciting the bending modes. Authors please make this distinction, possibly referring to the more general model of Hua Guo.

We've added the following sentence in the first paragraph of Page 6 to explain the reactivity enhancement by exciting the bending modes. "The reactivity enhancement by exciting the stretching modes is consistent with the Polanyi's rules for a late-barrier reaction, and the enhancement due to the bending mode is evidenced by the fact that the H–O–H angle of the saddle point (119.21o) is larger than that in isolated H₂O (104.41 o)."

5. Regarding the second paragraph on page 6: it would be good to include a Table stating not only the minimum (classical) barrier height in the PES, but also the zero-point energy corrected barrier height in the paper. This is crucial information. For instance, as the authors should also discuss in the paper, reaction below the classical barrier does not need to be indicative of tunneling effects, it could also be due to zero-point energy effects (if the zpe at the barrier is lower than the zpe of the reactants).

We've rewritten the begging of the second paragraph on Page 2 to now read "From the corresponding dissociation probabilities shown in a logarithmic scale in Fig. 2(b), we can see obviously that in the very low energy regions below the classical barrier (1.08 eV) there exist considerable dissociation probabilities for the ground vibrational state of H₂O. The zero-point energies (ZPEs) of reactant asymptote and transition state are 0.57 eV and 0.43 eV, respectively, so that the ZPE corrected barrier height is 0.94 eV. Thus, the reactivity below the classical barrier (down to roughly 0.9 eV) is due to both the ZPE and tunneling effects."

6. I suggest to extend Table 1 with one column, showing 6D results for the vibrational efficacy for one value of the reaction probability. This seems important in the light of the first paragraph on page 7, it would be better not to have to look up reference 46 to believe the statement made about the comparison between the 6D and 9D results.

We've added one more column in Table I to show the 6D results for the vibrational efficacy at $P=10^{-5}$ by Guo and co-workers.

7. I suggest that the authors change their terminology regarding the "two site approximations" on page 7, now called "site-averaging" and "site-sudden". Both approximations are actually of the "site-sudden", or "site-averaging type, they have just been implemented differently. Specifically, in one case the exact fitted potential is used, in the other case an approximation to it, using a harmonic expansion around one specific site. The terminology should be changed to reflect this, for instance "site-averaged exact potential" and "site averaged harmonic potential" with acronyms SAEP and SAHP. Also, I suggest to drop any mention of "two site approximations", it might be better to say "two implementations of site-averaging approximations", or something along these lines.

We are happy to completely accept the referee's suggestions about the terminology for the two site-averaging approximations. Changes have been made to the main text, SI, and also Figures. Now the two implementations of site-averaging approximations are site-averaging approximation with harmonic potential (SAHP) and site-averaging approximation with exact potential (SAEP).

8. Page 8, regarding the difference between "site-sudden" and "site-averaged": far away from the site around which the potential is expanded in a harmonic approximation, the expanded potential becomes too high. This was also invoked to explain AIMD and site-sudden results for methane + Ni(111), see Jackson et al., JCP141, 054102, 2014. The authors might discuss whether this problem is too expected in general if an harmonic expansion around a fixed site is used.

We've added the following sentence in the second paragraph of Page 9. "The potential expanded in a harmonic approximation around a fixed site can be too high, which is expected in general and was also discussed by Jackson *et al.*" And we also referred to the JCP141, 054102, 2014 paper.

9. In papers like these, the reaction probability is computed by calculating the probability that one of the OH bonds breaks. The correct reaction probability is only obtained if the computed "reactive flux" is multiplied by a factor two. If this was indeed done, this should be explicitly indicated in both the paper and the SI. If this was not yet done, all plots should be changed with reaction probabilities multiplied by a factor two, and again this should be indicated in the paper and the SI.

We've added the following sentence both in the main text and SI to make things clear. "Due to the non-reactive of one OH bond in current theory, the computed dissociation probabilities were multiplied by a factor of 2 which correspond to the results for two reactive bonds."

10. In the SI, I do not see the estimated convergence of the 9D QD calculations. Can the authors add results of convergence tests? This should be less difficult than it might seem. Convergence w.r.t. rotational basis set can be tested in calculations using a fixed site approximation. Convergence w.r.t. numbers of points in X and Y could be tested in a calculation where approximations are made to one or two of the rotations and/or the bond not allowed to be broken. The number of points in X and Y (7 x 7) would seem to be rather low in the 9D calculations, of the same order as in the "site-averaged" calculations (8 x 8 with 25 points). The question could then arise whether the 9D calculation represents a true benchmark.

The 9D results were converged quite well based on the present parameters. To verify this, we've added two figures, Fig. S2 and Fig. S3 in SI, which show the convergence of 7D dissociation probability at the TS site with respect to the rotational basis, and of 9D probability with respect to X and Y, respectively.

Minor points:

- a) Page 2: "...attracted to them."
Done!
- b) In line 9 of page 4 I suggest to talk of the work of "Guo and Beck and coworkers", i.e. mention both senior scientists on the paper.
Done!

- c) On page 4 I suggest to say in line 3 from the bottom: "The quantitatively accurate description of the title gas-surface reaction can only be verified by comparison with a fully coupled 9D ... ". In other words, approximative treatments might exhibit quantitative accuracy, but to be sure the benchmark full-dimensional (w.r.t. the molecular degrees of freedom) calculation needs to be done first.

Done! We' changed the sentence as the referee suggested.

- d) First two lines page 5: "... 9 degrees of freedom to be fully coupled."

Done!

- e) Reference 21 is wrong. It should be "Dai, JQ" and not "Dai, J". Authors please check all citations on the Web of Science, especially the ones with Chinese authors in them.

We've checked this reference and others, and do think they are correct.

- f) Regarding reference 22: the better paper to cite is Kroes, Baerends, Mowrey, Phys.Rev.Lett.78, 3583, 1997. This paper is the first 6D quantum dynamics paper on fully activated dissociation of H₂, and has both an earlier "received" and "accepted" date than Ref.21.

The paper (Phys.Rev.Lett. 78, 3583, 1997) has been added in the references too.

- g) A recommendation for future work is to use PBE instead of PW91. PBE was designed to supersede PW91, see the PRL 1996 of Perdew, Burke, and Enzerhof.

Thanks and we will use PW91 in future.

- h) The SI should state a measure of the accuracy with which the PES was fitted (for instance, a RMSD for data not in the training set).

We've added the following sentence in SI. "This fit results in an overall very small root mean square error (RMSE) of only 9.0 meV, but is significantly smaller (6.0 meV) for energy points below 2.0 eV relative to the H₂O + Cu(111) asymptote, representing the unprecedented fitting accuracy for PESs of polyatomic-surface reactions."

- i) In the SI on page 3, line 5 should mention that the expansion is also in Fourier basis functions of X and Y.
Done!
- j) Page 5 SI: line 5 paragraph 3: should read "top, hcp, and fcc sites". I.e., replace "hollow" by "hcp". Same elsewhere.
Done!
- k) Bottom page 6 SI: the readability of the SI can be tremendously improved by making reference to Fig.4a of the main paper for the numbering of the sites. Authors: many readers will read the paper in the order Intro, SI, rest of paper.
We did indicate at that place in SI: All the sites are labeled and presented in Fig. 4(a).
- l) The caption of Fig.S3 should likewise refer to Fig.4a of the main paper.
Done!

Sincerely yours,

Professor Dong Hui Zhang
State Key Laboratory of Molecular Reaction Dynamics
Dalian Institute of Chemical Physics, CAS
Dalian, China 116023
Tel: (86)-411-84379362
Fax: (86)-411-84675584
E-mail: zhangdh@dicp.ac.cn

Reviewer #3 (Remarks to the Author):

The authors have dealt with my comments to my complete satisfaction. As I already said in my first report, their work represents a major achievement, and is fully publishable in Nature Communications. I therefore am happy to recommend publication of the revised manuscript.

REVIEWERS' COMMENTS:

Reviewer #3 (Remarks to the Author):

The authors have dealt with my comments to my complete satisfaction. As I already said in my first report, their work represents a major achievement, and is fully publishable in Nature Communications. I therefore am happy to recommend publication of the revised manuscript.

Finally, we thank the referees for their useful comments and recommendations of publication of the manuscript.